# Hepatitis E Virus in Finland: Epidemiology and Risk in Blood Donors and in the General Population

**DOI:** 10.3390/pathogens12030484

**Published:** 2023-03-18

**Authors:** Jaana Mättö, Niina Putkuri, Ruska Rimhanen-Finne, Päivi Laurila, Jonna Clancy, Jarkko Ihalainen, Susanne Ekblom-Kullberg

**Affiliations:** 1Finnish Red Cross Blood Service, 01730 Vantaa, Finlandsusanne.ekblom-kullberg@veripalvelu.fi (S.E.-K.); 2Finnish Institute for Health and Welfare, 00100 Helsinki, Finland; 3Finnish Red Cross Blood Service Biobank, 01730 Vantaa, Finland

**Keywords:** hepatitis E virus, transfusion-transmitted infection, blood donor, HEV prevalence, seroprevalence, risk assessment

## Abstract

Autochthonous hepatitis E (HEV) cases have been increasingly recognized and reported in Europe, caused predominantly by the zoonotic HEV genotype 3. The clinical picture is highly variable, from asymptomatic to acute severe or prolonged hepatitis in immunocompromised patients. The main route of transmission to humans in Europe is the ingestion of undercooked pork meat. Transfusion-transmitted HEV infections have also been reported. The aim of the study was to determine the HEV epidemiology and risk in the Finnish blood donor population. A total of 23,137 samples from Finnish blood donors were screened for HEV RNA from individual samples and 1012 samples for HEV antibodies. Additionally, laboratory-confirmed hepatitis E cases in 2016–2022 were extracted from national surveillance data. The HEV RNA prevalence data was used to estimate the risk of transfusion transmission of HEV in the Finnish blood transfusion setting. Four HEV RNA-positive were found, resulting in 1:5784 (0.02%) RNA prevalence. All HEV RNA-positive samples were IgM-negative, and genotyped samples represented genotype HEV 3c. HEV IgG seroprevalence was 7.4%. From the HEV RNA rate found in this study and data on blood component usage in Finland in 2020, the risk estimate for a severe transfusion-transmitted HEV infection is 1:1,377,000 components or one in every 6–7 years. In conclusion, the results indicate that the risk of transfusion-transmitted HEV (HEV TTI) in Finland is low. However, continuous follow-up of the HEV epidemiology in relation to the transfusion risk landscape in Finland is necessary, as well as promoting awareness in the medical community of the small risk for HEV TTI, especially for immunocompromised patients.

## 1. Introduction

Hepatitis E virus (HEV) is a major cause of enterically transmitted hepatitis in developing countries, where it is transmitted by the fecal–oral route like hepatitis A [1]. During the last two decades, HEV has become an emerging cause of viral hepatitis in Western industrialized countries as an increasing number of autochthonous HEV infections has been recognized in many European countries [2,3,4].

There are four major HEV genotypes known to infect humans with differing epidemiology. Genotypes 1 and 2 are waterborne and causative of epidemics among humans in many tropical countries. Zoonotic reservoirs for genotypes 1 and 2 have not been reported. Genotype 3 and 4 infections are zoonotic diseases infecting humans and mammals and are found mainly in industrialized nations [1,2,3,5]. Further, HEV genotype 7 has been identified in camels and has also been associated with chronic hepatitis in an immunocompromised liver transplant patient [6,7]. HEV G3 is the prevalent genotype in Europe, where pigs are the principal animal reservoir, and the main route of transmission to humans is the ingestion of undercooked pork meat [2,5]. In addition, close contact with pigs and other animal reservoirs has been associated with higher anti-HEV seroprevalence than in the general population in several studies. For example, data on HEV seroprevalence from France and Germany indicate professional HEV exposure in swine farm workers [8,9]. Infections from other meat sources (e.g., wild boar and deer) and shellfish have been reported, and the virus has been found in many typical sources for enteric hepatitis infections, like water and vegetables. The role of these sources in viral transmission is unclear [1,5].

Parenteral transmission of HEV infections via blood transfusion has been documented in Japan and several European countries since the mid-2000s [2,5,10,11]. In addition, Australia reported one transfusion-transmitted HEV infection in 2017 [12]. Most HEV G3 infections are asymptomatic or cause only mild self-limiting disease in immunocompetent individuals and remain unrecognized. Thus, most of the infections in healthy blood donors are subclinical or symptom-free [13]. However, in patients with pre-existing liver disease and in immunocompromised patients, e.g., patients who have received organ or stem cell transplants, the clinical course can be severe or prolonged, leading to chronic hepatitis and, eventually, cirrhosis [14]. For these patients, treatment with ribavirin or a reduction of immunosuppression has been successful in achieving HEV RNA clearance, but also some fatal cases have been reported [5]. Further, HEV transmission is not effectively prevented by pathogen reduction methods [15].

Several European countries have implemented screening of blood donors for HEV RNA based on a national risk assessment [14,16,17,18,19,20,21,22,23,24]. There are also countries that have decided not to implement screening, e.g., Denmark, where transmission risk in a study in 2015 was found to be low [25]. Generally, the main factors influencing the risk are the rate of HEV viraemia among blood donors and the viral load in the blood components, together with patient-related risks such as multiple transfusions and immunosuppression [11,14].

There is only limited data concerning HEV epidemiology in the Finnish population [3,26]. In 2009, HEV seroprevalence was 10.2% among Finnish veterinarians participating in the national veterinary congress [27]. The first domestic hepatitis E case was reported in 2013 [28]. The epidemiology of HEV among blood donors in Finland was only studied in a very limited group 30 years ago [29]. No transfusion-transmitted HEV infection has been reported in Finland.

The aim of the study was to determine the prevalence and incidence of HEV infection in the blood donor population in Finland. Further, the aim was to apply the results to estimate the risk of transfusion transmission of HEV infection in Finland. Data from the Finnish Infectious Disease Registry (FIDR) on nationally notified HEV infections in 2016–2022 are reported to describe the epidemiology of hepatitis E in Finland. The FIDR performs surveillance of infections based on reports from clinical laboratories and healthcare providers. Notifications of HEV infections to the FIDR are mandatory, but this does not preclude underreporting caused by underdiagnosis. Collectively, the data can be applied as a basis for risk-based decisions on possible screening of blood donations for HEV RNA.

Combining the blood donor HEV RNA prevalence data and FIDR registry data, which both cover datasets from all regions in the country, provides valuable information on HEV epidemiology in Finland. The information has been lacking thus far, which has hindered the evaluation of the HEV TTI risk in Finland. Our results also improve the assessment of the national public health impact of HEV.

## 2. Materials and Methods

The study material comprises blood donor samples and national surveillance data on laboratory-confirmed HEV cases.

### 2.1. Blood Donor Samples

A total of 23,199 blood donor samples were obtained from all ten regional donation sites in Finland. The 3.5 mL EDTA blood samples were collected during standard whole blood donations from March 2020 to March 2021 from donors who previously had given written biobank consent for the Finnish Red Cross Blood Service Biobank Vantaa, Finland. After centrifugation at 2500× *g* for 8 min, the samples were aliquoted and pseudonymized by the biobank. In addition to the samples, demographic data, including gender, age, and donation site, were provided by the biobank for the study. The coverage in relation to the number of blood donations during the same period in each location is presented in Table 1. As the samples used in the study were obtained from donors who had previously given biobank consent, samples from first-time donors were not available for the study. There is a slight overrepresentation of regular, older blood donors compared to the entire Finnish donor pool. This is not considered to be of significance for assessing the epidemiology of HEV in our donor population, as Biobank donors are well-represented in all age groups. Sixty-two samples could not be analyzed due to technical reasons.

A written biobank consent was obtained from all the donors, and the donors were further informed about the HEV prevalence study. The use of the samples and data is in accordance with the biobank consent and meets the requirements of the Finnish Biobank Act 688/2012. The study was approved by the Ethics committee of the Helsinki University Hospital (HUS/3415/2019).

### 2.2. National Data on Hepatitis E in the Finnish Infectious Disease Registry

In addition to the blood donor samples, surveillance data from FIDR was used in the study. Clinical microbiology laboratories in Finland are obliged to report laboratory-confirmed hepatitis E cases diagnosed by serology or PCR to FIDR. From the beginning of 2016, the laboratories have been encouraged to notify only anti-HEV IgM-positive cases.

A database was formed of hepatitis E notifications from 1 January 2016 to 31 December 2022. The cases were characterized according to gender, age, hospital district, travel history and death. Incidence rate ratio (IRR) was calculated for 10-year age groups. Information on IgM status was obtained from notification footnotes. Genotyping results of sixty-nine serum samples collected as part of HEV outbreak investigations in Finland during years 2019–2022 are included in the present study.

### 2.3. HEV RNA Screening of Blood Donor Samples with Procleix HEV Assay

In total, 23,137 blood donor samples were screened for the presence of HEV RNA. HEV RNA analysis was performed as individual testing (ID NAT) from fresh EDTA plasma samples. Transcription-mediated amplification (TMA) based Procleix HEV assay implemented on the fully automated Procleix Panther System (Grifols Diagnostics Solutions Inc., Barcelona, Spain) was applied for HEV RNA analysis. Procleix HEV assay is a single-tube assay with sample preparation with target capture, target amplification by transcription-mediated amplification, and detection of the amplification products by the hybridization assay. The test run is valid if positive and negative calibrator results are verified and accepted by the Procleix Panther System Software 5.2. In addition, an internal control was added to each reaction to evaluate sample validity. Testing was performed according to the manufacturer’s instructions. Samples showing reactive results were re-tested once or in duplicate. Samples with two reactive test results were defined as HEV RNA-positive. Before analysis of the samples, the manufacturer’s performance specifications were verified.

### 2.4. Anti-HEV IgG and Anti-HEV IgM Analysis of Blood Donor Samples

In total, 1009 randomly selected samples and the three repeatedly HEV RNA reactive samples were screened for the presence of HEV-specific IgG antibodies. Testing was performed with the VIDAS anti-HEV IgG assay (BioMérieux, Marcy-L’Etoile, France). Anti-HEV IgG-positive samples were retested. Further, 71 anti-HEV IgG-positive samples, 11 initially reactive HEV RNA samples and three HEV RNA-positive samples (altogether 88 samples) were analyzed with VIDAS anti-HEV IgM assay (BioMérieux, Marcy-L’Etoile, France) to evaluate the prevalence rate of recent infection. Anti-HEV IgG and anti-HEV-IgM analyses were performed according to the manufacturer’s instructions for the assays.

### 2.5. HEV RNA Analysis with the Confirmatory PCR Test and HEV Genotyping

Fourteen blood donor samples, including two repeatedly reactive and 12 initially reactive samples in the HEV RNA screening with Procleix HEV assay, were re-tested with a confirmatory PCR test at the Finnish Institute for Health and Welfare. One repeatedly reactive sample could not be analyzed due to inadequate sample volume. Samples were concentrated before RNA extraction by centrifugation (25,000× *g* for 1 h at +4 °C). Supernatants were added to 300 µL PBS and incubated at room temperature for 10 min. Viral RNA was extracted using Chemagic Viral300 DNA/RNA Kit and Chemagic™ 360 instrument (PerkinElmer, Waltham, MA, USA) according to the manufacturer’s instructions.

The national surveillance data includes analyses of the HEV RNA and HEV genotypes of sixty-nine serum samples from HEV outbreak investigations in 2019–2022. Viral HEV RNA was extracted from the 2021–2022 outbreak samples (34 samples) by the method described above for the blood donor samples, and from the 2019–2020 outbreak samples (35 samples) using NucliSens Magnetic Extraction Reagents (Biomerieux, Marcy-l’Étoile, France) according to the manufacturer’s instructions.

HEV-positive RNA extracts were sequenced (493 nt). Genotyping primers and cDNA programs are described in Boxman et al. 2017 [30]. cDNA reaction mix, PCR reactions were performed according to Nix et al. 2006 [31]. PCR products were purified using QIAquick PCR Purification Kit (Qiagen) according to manufacturer’s instructions. Sequencing reactions with BigDye Terminator v3.1 cycle sequencing kit (Applied Biosystems) and sequencing with ABI3730xl Automatic DNA Sequencer (Applied Biosystems) were performed by Institute for Molecular Medicine Finland (FIMM) Sequencing Laboratory. The electropherograms were analyzed using Sequencher 5.4.6 software (Gene Codes Corporation). Samples were genotyped using Hepatitis E Virus Genotyping Tool Version 0.1, RIVM (https://www.rivm.nl/mpf/typingtool/hev/) (accessed on 16 March 2023). Multiple sequence alignments were made with MEGA-X [32]. The phylogenetic tree was estimated with MEGA-X using Neighbor-joining model [33]. Phylogenetic analyses were performed on HEV sequences derived from blood donors and compared with outbreak investigation samples and human HEV strains from GeneBank. Sequenced viruses from blood donors have been submitted to GenBank with accession numbers OQ596308, OQ596309, and OQ596310.

### 2.6. Risk Assessment Modelling

The HEV RNA rate found in our study was used to estimate the risk of transfusion transmission of HEV and severe adverse outcomes in a recipient by a risk assessment model described in Hoad et al. 2017 [34]. Transmission risk data is based on the results from a large HEV transfusion transmission study in England reported by Hewitt et al. 2014 [11]. According to the Finnish Red Cross Blood Services’ (FRCBS) data on issued blood components and blood transfusions, the usage of issued labile components in Finland is high; follow-up data by FRCBS have shown that 99% of issued red cell components and 90% of issued platelet components are transfused. In Finland, the pharmaceutical SD plasma OctaplasLG product is used for plasma transfusions, and thus plasma components are not included in the risk model.

The modified risk model was performed with the following assumptions:(1)The HEV RNA prevalence in fresh components is the same as that in whole blood donors,(2)A total of 42% of HEV RNA-positive donations result in a transfusion transmission infection (viremia, seroconversion) [11],(3)A total of 5% of the HEV-infected recipients will have an adverse outcome (symptomatic infection) [11],(4)A total of 1% of the HEV-infected recipients have a severe adverse outcome (e.g., untreatable chronic infection) [11],(5)Background immunity in the recipient population is not considered,(6)A total of 99% of issued red cells and 90% of issued platelet components are transfused.

The risk estimates for HEV TTI are calculated and reported as the numbers and proportion of components leading to adverse and severe adverse events in relation to components transfused in Finland in one year.

## 3. Results

### 3.1. HEV RNA in Blood Donors

In the HEV RNA analysis, 3 out of 23,137 (0.012%) blood donor samples were repeatedly reactive with the Procleix Panther HEV assay. In addition, 13 (0.06%) samples were initially reactive (positive only in one out of three repeats) and showed a S/CO value ranging from 1.14 to 10.44 in the test (Table 2). The two repeatedly reactive samples tested with the secondary PCR test were positive (one sample could not be tested due to insufficient sample volume). In addition, one of the initially reactive samples showing the highest S/CO value in the Procleix Panther HEV test, S/CO being 10.44, was reactive in the secondary PCR test. The 12 samples showing initial reactive results with the Procleix HEV assay tested negative with confirmatory PCR were considered false-positive samples. Three samples showing a reactive result in the secondary PCR were genotyped (one sample could not be tested due to insufficient sample volume) and represented all the HEV 3c genotypes (Table 2).

Three out of four HEV RNA-positive samples were collected from donations in the HYKS region encompassing the Helsinki Metropolitan area in Southern Finland and one in the KYS region in Eastern Finland (Figure 1). Three HEV RNA-positive samples were from female blood donors (one each in the age groups 30–39, 40–49, and 50–59 years) and one from a male donor (in age group 18–29 years).

Collectively, four samples were interpreted as HEV RNA-positive, resulting in a HEV-RNA prevalence of 1:5784 (0.02%, 95% CI 0.005–0.044) or 17 HEV RNA-positive samples per 100,000 donations among Finnish blood donors.

### 3.2. HEV Seroprevalence in Blood Donors

Of the 1012 blood donor samples analyzed for anti-HEV IgG, 75 were positive (7.4%, 95% CI 5.9%–9.2%, Table 3). Seroprevalence ranged from 3.3% in the group of 18–29-year-old males to 11.9% in the group of 30–39-year-old males (due to a low number of samples studied in age groups 18–19 (3) and 70 (13), these were attached to age group 20–29 and 60–69, respectively). Seroprevalence was on a similar level between males and females (7.6% vs. 7.2%).

Regional distributions of the anti-HEV IgG-positive samples and HEV RNA-positive samples are presented in Table 4 and the corresponding seroprevalence in Figure 1a. Seroprevalence varied from 4.0% to 9.7%. The coverage of sample collection between geographic regions was good except for the OYS region (Northern Finland).

Four out of 88 samples analyzed for anti-HEV IgM were positive. All anti-HEV IgM-positive samples were also anti-HEV IgG-positive, and none of the IgM-positive samples were HEV RNA-positive (Table 2). Three anti-HEV-IgM-positive samples were obtained from females and one from a male, and the donors were 35–68 years old. Only one IR/RR HEV RNA sample was anti-HEV IgG-positive, showing a weak reaction in two out of three test repeats (values being 0.65 and 0.78 U/mL) (Table 2).

### 3.3. HEV Notifications in Finnish Infectious Disease Registry

Between 2016 and 2022, 249 hepatitis E cases were reported to FIDR (22–57/year; Figure 2). The mean annual incidence was 0.7:100,000 inhabitants (0.4 to 1.0:100,000/year). The incidence decreased from 1.0 to 0.5:100,000 in 2019–2022. The median age of cases was 58 years (range 4–88 years). Males (147/249; 59%) were more often reported than females, but there was no significant difference between the incidence rates (IRR 1.4; 95% CI 0.7–3.0, *p* = 0.293). The incidence was almost 3-fold in over 50-year-olds compared to those under 50 years (IRR 2.7; 95% CI 1.3–6.0, *p* = 0.003) (Figure 3). Cases were reported from all except one hospital region (Åland), with mean annual incidence ranging from 0.5–0.8 per 100,000 (Figure 1b). Among the samples tested, the main genotype was HEV-3e (23/37; 62%), followed by HEV-3c (12/37; 32%) (Figure 2).

Of 249 cases, 111 (45%) had not travelled abroad within the incubation period, while for 114 (46%), the information on travelling abroad was missing. Of the cases, five (2%) died within 30 days of the sampling date. Additional information on the cases was given in the footnotes of 94 (38%) notifications. IgM status was not mentioned for 50 cases, while 36 (82%) cases were reported to be anti-HEV IgM-positive and 8 (18%) anti-HEV IgM-negative cases.

### 3.4. Assessment of Blood Transfusion-transmitted HEV Infection

Based on the risk of a viremic donation (0.017%, 95% CI 0.005–0.044%) found in our study and the risk calculation model used [34], the estimated risk of a symptomatic infection per component transfused is 1:275,440 and the risk of a severe infection 1:1,377,202 (Table 5). With an annual transfusion of 200,000 labile blood components, this means one severe case in approximately 7 years. The confidence intervals for the risk estimates are substantial (Table 5), which must be considered when evaluating the results.

## 4. Discussion

HEV prevalence in blood donors has been studied in numerous countries worldwide by assessing the rates of anti-HEV IgG, anti-HEV IgM and/or HEV RNA positivity [10,14,16,35]. Data on the HEV prevalence in Finnish blood donors have been missing thus far. This information is needed for evaluating the risk of HEV transmission via blood transfusion and, subsequently, the strategy for blood donor HEV testing in Finland.

We found a low rate of HEV viremia among blood donors in Finland. Four samples were confirmed HEV RNA-positive, resulting in a HEV RNA prevalence of 1:5784 (0.017%) in Finnish blood donors. The rate of HEV RNA positivity among blood donors varies greatly between countries worldwide, and variation is also high between European countries from 1:744 to 1:8636 [10,14,35]. The prevalence observed in our study is among the countries with fairly low rates, such as Ireland 1:4997 [19], Spain 1:4341 [24], Belgium 1:5448 [36], and Austria 1:8416 [37]. The observed rate in our study is below the overall EU rate of 1:3109 donations [16].

We performed HEV RNA screening from individual samples by applying the Procleix HEV assay. Blood donor screening is often performed from mini-pool samples, and in several prevalence studies, pooled samples have been used, which has an impact on the sensitivity of testing. We were able to genotype three HEV RNA-positive samples, and they all represented HEV genotype 3c, which along with subtypes 3e and 3f, is one of the most frequently reported subtypes in the EU [16]. In Finland, HEV-3e was the most common genotype, followed by HEV-3c in cases reported to FIDR 2019–2022. The genotype data, and the surveillance data on the country of infection origin, suggests that the infections are of domestic or European origin. HEV genotype 3 is endemic in Europe and is transmitted mainly zoonotically; HEV-3e and HEV-3c have been found in both humans and pigs [3]. The method applied for HEV RNA analysis was qualitative, and due to the limited volume of samples available, we were not able to analyze the viral load in the HEV RNA-positive samples.

In our study, the VIDAS anti-HEV IgG assay was applied for the seroprevalence assessment, while the most used assay in the blood donor seroprevalence studies was the Wantai Elisa test. The analytical and clinical performance of the VIDAS assay has been shown to be excellent [38], and although some discrepancies in anti-HEV IgG results of individual samples have been seen in comparative studies, the agreement between the VIDAS and Wantai anti-HEV IgG assays has been reported to be good [39].

The observed anti-HEV IgG seroprevalence in Finnish blood donors was 7.4%. Similar to the rates of HEV RNA positivity, HEV seroprevalence in blood donors has been studied in numerous countries in Europe and worldwide, and there is a large variation in the anti-HEV IgG rates between the countries ranging from 4.7% to 43.5% [14]. The seroprevalence in Finland compares to the lower end reported in European countries. We observed slightly higher rates of anti-HEV IgG positivity towards the older blood donor age groups, although the highest rate was observed in the group of 30–39-year-old males. Furthermore, in the HEV cases reported to the FIDR, the higher incidence in the older age group is observed, the incidence being the highest among 60–69-year-old males. In several previous studies, differences have varied from the highest rate detected in young donors [19] to higher rates in the older age groups [23] or in males [24]. The findings suggest that age and gender differences are linked to the national epidemiological situation.

An increase in hepatitis E reporting has been seen in Western Europe in recent years [3]. However, HEV is not notifiable at the European level, and no EU-wide case definition exists, indicating that surveillance systems and testing recommendations may vary across the countries. During 2016–2022, the incidence of hepatitis E in Finland was stable, except in 2019, when an outbreak of HEV-3e was identified, with nine cases in several municipalities in Finland. The investigation was discontinued while resources were directed to COVID-19 pandemic management, but several common meat and vegetable exposures were identified among the interviewed cases. In 2022, seven cases with similar HEV-3e and meat and vegetable exposures were identified.

HEV infection in a healthy adult population is most often subclinical, as demonstrated by the RNA-positive blood donors in our study and other research populations. In our national data, the case fatality rate was 2%. Mortality rates up to 10% have been reported [40], but the burden of HEV infection in humans in Europe is scarcely documented. More serious forms of HEV affect immunocompromised patients and those with serious liver disease. Unfortunately, these patients also need many transfusions [5,11]. Since the reporting of infections from diagnostic laboratories is mandatory, the FIDR data provides a trustworthy overview of the infections with the classical form of the disease, and up to now, no blood transfusion-related infections have been reported. In addition to the hepatic manifestations, immunological symptoms like Guillain-Barré syndrome have been linked to HEV [10]. Whereas HEV is mostly included in the diagnostic setup of hepatitis in immunocompromised patients, atypical disease forms may go undiagnosed.

We assessed the risk of transfusion-transmitted HEV infection in Finland by applying the risk assessment model described by Hoad et al. [34] for evaluating the risk in Australian blood donors. Based on the HEV RNA positivity rate and the annual number of components transfused, the estimated risk of a symptomatic TT HEV infection case is one in 1.4 years, and for severe infection, one in seven years. In the Australian study, far lower risk was observed as the HEV RNA prevalence in Australia is 1:74,131, currently, the lowest prevalence reported globally [34]. Since the primary route of HEV transmission in Europe is food and especially undercooked pork meat, also dietary factors, contact with animals and other related factors should be considered in evaluating the HEV infection risk, especially for patients with increased risk for a chronic infection. In our study, we could not include information on dietary exposure in the risk assessment. HEV infection has been reported to be very common among Finnish piglets, but the risk of foodborne transmission from the piglets has been determined to be low since only a few have been shown to carry the virus at slaughter age [26]. Antibodies for the hepatitis E virus have been found in 18% of Finnish wild boars hunted in 2016 [41], and seroprevalences of 9% and 1.4% were reported in Finnish moose, and white-tailed deer hunted in 2008–2009 [42]. Still, the foodborne infection route needs to be taken into consideration in Finland. In a study by Tedder et al. [43], the risk between dietary exposure versus unscreened transfusion-transmitted HEV in the UK was evaluated, and besides the group of immunosuppressed patients receiving extensive amounts of blood components for a long period of time, the dietary exposure is higher than the transfusion transmission risk.

Several blood establishments in Europe have implemented universal or selective HEV screening of blood donations to mitigate the risk of HEV TTI (Table 6).

Although the national blood donor prevalence data is considered in planning the screening strategy of a given country, we observed that the screening strategies do not systematically follow the epidemiological factors. HEV RNA prevalence rates in the countries applying universal HEV screening vary from 1:845 to 1:4745 [16], while in some countries, e.g., Denmark [25] and Poland [53] with higher HEV RNA detection rates than the overall EU rate 1:3109 [16], HEV screening is not applied, suggesting that several factors are considered in the national level risk assessment. Changes in the prevalence rates over time are reported. Fluctuations in the UK HEV RNA positivity rate have been reported, varying from 1:1365 in earlier studies to 1:3830 in a study reporting the outcome of the universal screening during 2016–2017 and even lower (1:4781) during the time of reporting of the study by Harvala et al. [17]. In Catalonia, Spain, the HEV RNA positivity rate among blood donors was 1:4341 during universal screening in 2017–2020 and 1:3333 in 2013 [24,54].

## 5. Conclusions

There appears to be a correlation between the rates of HEV IgG reactivity and RNA positivity of blood donors in different populations (Table 6). The screening policies do not exactly follow the epidemiology, as demonstrated in Figure 4. The countries having higher RNA positivity rates, as well as a higher prevalence of IgG antibodies against HEV, have been more active in implementing routine screening, but there are exceptions at both ends of the epidemiological spectrum. This demonstrates the significance of national risk management prioritizations.

The prevalence of HEV RNA positivity and anti-HEV IgG among Finnish blood donors are relatively low in comparison to prevalence in European countries in general. Moreover, there appears to be no significant increasing trend in the infection rate among the general population. Studies conducted at slaughterhouses demonstrate a low level of risk from domestic meat products [57,58]. However, the food-derived risks from different sources can be estimated to be much higher than the risks from parenteral transfusion at the population level.

So far, routine HEV screening of blood donors is not applied in Finland. The results of the present study indicate that the estimated risk of HEV TTI in Finland is low. Continuous follow-up and re-evaluation of HEV as a part of the transfusion risk landscape are necessary. In addition, the medical community in Finland should be aware of a small risk of HEV TTI in addition to the risk for autochthonous foodborne HEV infection, especially for immunocompromised patients. Data reported in the present study provide new information on HEV epidemiology in Finland, which is valuable for evaluating and managing the HEV TTI risk in Finland. The approach applied in the present study, where blood donor prevalence data and national epidemiological registry data are combined, is current and improves the assessment of the national public health impact of HEV.

## Figures and Tables

**Figure 1 pathogens-12-00484-f001:**
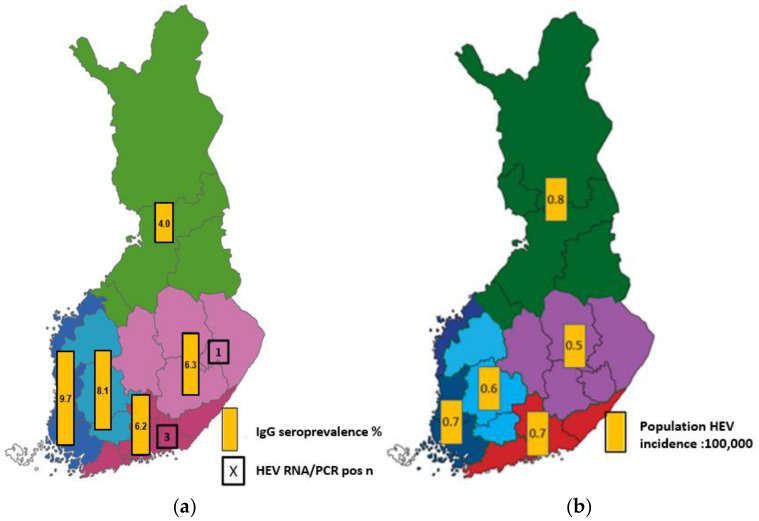
(**a**) Regional distribution of anti-HEV IgG-positive and HEV RNA-positive blood donor samples (%) (chapters 3.1. and 3.2.). (**b**) Mean regional HEV incidence (:100,000) based on the cases reported to the Finnish Infections Disease Registry during 2016–2021 (chapter 3.3). Regions: HYKS (red), TYKS (dark blue), TAYS (light blue), KYS (purple), OYS (green).

**Figure 2 pathogens-12-00484-f002:**
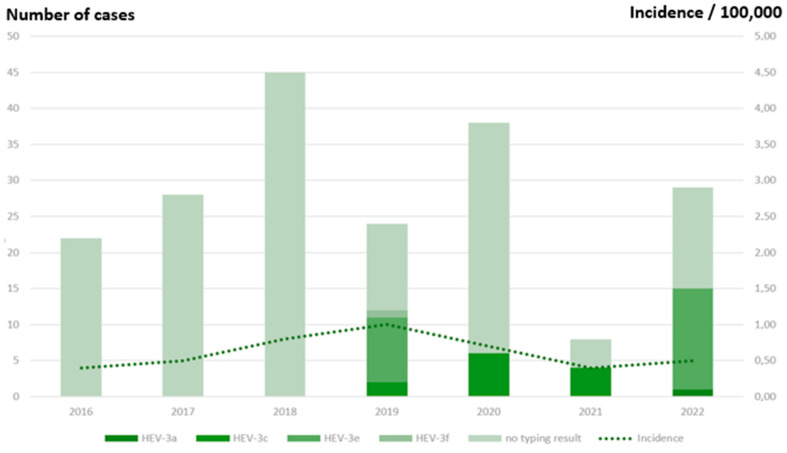
Annual number of HEV cases reported to the Finnish Infectious Disease Registry according to the typing result and incidence per 100,000 inhabitants, 2016–2022.

**Figure 3 pathogens-12-00484-f003:**
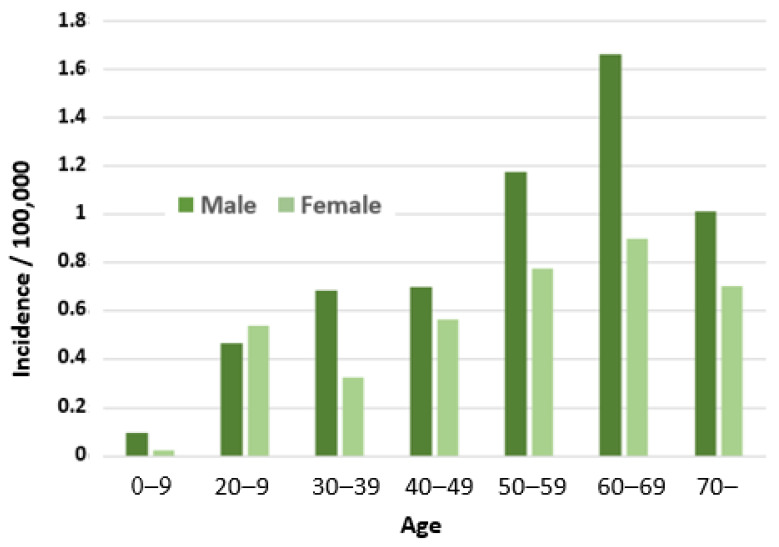
Incidence of HEV (:100,000) in different age and gender groups based on the HEV cases (total N = 249) reported to the Finnish Infectious Disease Registry during 2016–2022.

**Figure 4 pathogens-12-00484-f004:**
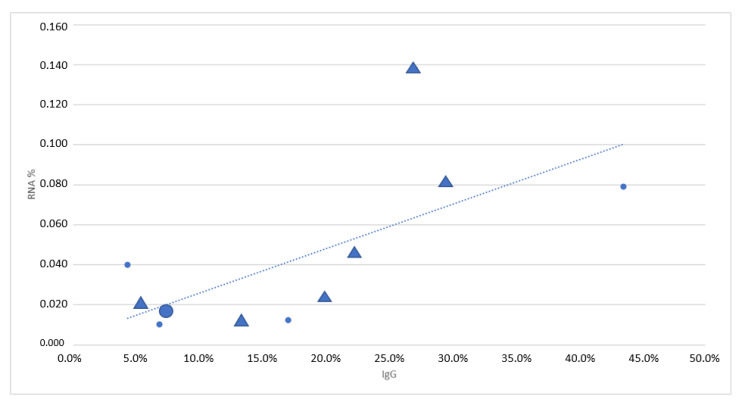
The HEV RNA and IgG frequencies of countries doing routine blood donor screening are marked with triangles (Ireland, Spain, The Netherlands, France, and Germany] and those not doing routine screening are marked with small circles (Italy, Sweden, Denmark, and Poland). Our result has been marked with a bigger circle.

**Table 1 pathogens-12-00484-t001:** Regional distribution of the population in Finland, HEV incidence based on the cases reported in the Finnish Infections Disease Registry (FIDR) and distribution of blood donations and the samples collected for this study.

Healthcare Region *	Population in Finland (31 December 2020)	HEV Cases in FIDR Year 2020	HEV Incidence/100,000	Blood Donations 3/2020–3/2021	Blood Donor HEV Samples **	Blood Donor Anti-HEV IgG Samples **
HYKS	2,198,182	16	0.7	82,739 (37%)	9138 (39%)	386 (38%)
TYKS	869,004	12	1.4	36,590 (17%)	5762 (25%)	186 (18%)
TAYS	902,681	4	0.4	37,693 (17%)	4304 (19%)	268 (26%)
KYS	797,234	3	0.4	25,389 (11%)	3551 (15%)	143 (14%)
OYS	736,563	3	0.4	37,664 (17%)	355 (2%)	25 (2%)
Åland	30,129	0	0	956 (0.4%)	84 (0.4%)	3 (0.3%)
Total	5,533,793	38	0.7	221,031	23,199	1012

* HYKS—Helsinki and Uusimaa University Hospital, TYKS—SouthWest University Hospital, TAYS—Tampere University Hospital, KYS—Kuopio University Hospital, OYS—Oulu University Hospital. Locations on the map of Finland are presented in Figure 1. ** In total, 23,199 samples were obtained and 23,137 samples studied for HEV RNA. Collection location for five HEV RNA samples and one HEV IgG sample unknown. Proportion (%) of all donations/samples collected during the time period of the study.

**Table 2 pathogens-12-00484-t002:** Hepatitis E RNA (HEV RNA) positive and initially reactive samples.

Sample	Collection Time	HEV RNA (Procleix NAT)		HEV RNA (PCR)	Geno-Type	Serology	
		Interpretation	Signal Per Cut-Off			Anti-HEV IgG	Anti-HEV IgM
HEV RNA-positive samples							
1	July 2020	RR	11.66/10.42	na	na	negative	negative
2	Aug 2020	RR	58.51/40.83	positive	3c	negative	negative
3	Sept 2020	RR	32.14/31.85	positive	3c	negative	negative
4	July 2020	IR	10.44/0.0	positive	3c	negative	negative
HEV RNA initially reactive samples							
5	July 2020	IR	1.7/0.0/0.07	negative	na	positive (2/3)	negative
6	July 2020	IR	8.44/0.0	negative	na	negative	negative
7	July 2020	IR	2.32/0.0	negative	na	negative	negative
8	July 2020	IR	1.99/0.0	negative	na	negative	negative
9	Aug 2020	IR	6.99/0.03/0.0	negative	na	negative	negative
10	Sept 2020	IR	1.76/0.0	negative	na	negative	negative
11	Oct 2020	IR	2.09/0.0	negative	na	negative	negative
12	Oct 2020	IR	1.64/0.0	negative	na	negative	negative
13	Jan 2021	IR	1.14/0.14	negative	na	negative	negative
14	Jan 2021	IR	1.16/0.06/0.0	negative	na	negative	negative
15	Feb 2021	IR	3.88/0.0/0.0	negative	na	negative	negative
16	Mar 2021	IR	1.67/0.08/0.0	na	na	na	na

**Table 3 pathogens-12-00484-t003:** IgG seroprevalence in different age and gender groups.

Age Group	Anti-HEV IgG: All				Anti-HEV IgG: Males			Anti-HEV IgG: Females		
	*n* Total	n Positive	% Positive	95% CI	*n* Total	*n* Positive	% Positive	*n* Total	*n* Positive	% Positive
18–29	170	7	4.1	1.7–8.3	61	2	3.3	109	5	4.6
30–39	146	12	8.2	4.3–13.9	59	7	11.9	87	5	5.7
40–49	197	8	4.1	1.8–7.8	97	4	4.1	100	4	4.0
50–59	244	22	9.0	5.7–13.3	139	12	8.6	105	10	9.5
60–70	255	26	10.2	6.8–14.6	115	11	9.6	140	15	10.7
Total	1012	75	7.4	5.9–9.2	471	36	7.6	541	39	7.2

**Table 4 pathogens-12-00484-t004:** Anti-HEV IgG seroprevalence by sampling region.

Region	Anti-HEV IgG		
	n Total	n Positive	% Positive (95% CI)
HYKS	386	24	6.2 (4.0–9.1)
TYKS	268	26	9.7 (6.4–13.9)
TAYS	186	15	8.1 (4.6–12.9)
KYS	143	9	6.3 (2.9–11.6)
OYS	25	1	4.0 (0.1–20.3)
Åland	3	0	0.0
Total	1012 *	75	7.4 (5.9–9.2)

* Collection location for one HEV IgG sample unknown.

**Table 5 pathogens-12-00484-t005:** Estimation of the risk of HEV transfusion transmission and adverse outcomes per component transfused in one year in Finland (modified from the model by Hoad et al., 2017 [34]).

	Blood Component			Risk of One Adverse Event Per x Component Transfused (95% CI)
	Red Cell Components	Platelets *	Total n (95% CI)	
Number of components issued in Finland 2020	179,387	31,381	210,768	
Number (proportion) transfused	177,593 (99%)	28,243 (90%)	205,836	
A. Risk of viremia (1:5784 = 0.017% of donations; 95%CI 0.005–0.044%) in x components	30.7	4.9	35.6 (9.7–91.2)	
B. Number of transmissions resulting in infection; estimated 42% of viremic products (0.42 * A)	12.9	2.1	14.9 (4.1–38.3)	1:13,772 (1:50,204–1:5375)
C. Number of symptomatic infections = 5% (0.05 * B)	0.6	0.1	0.7 (0.2–1.9)	1:275,440 (1:1,013,171–1:107,492)
D. Number of severe infections = 1% (0.01 * B)	0.13	0.02	0.15 (0.04–0.38)	1: 1,377,202 (1:5,065,856–1:537,461)
Risk of one symptomatic infection in x years **	1 in 1 year (95% CI 1 in 5 years to 1 in 0.5 years)			
Risk of one severe infection in x years **	1 in 7 years (95% CI 1 in 25 years to 1 in 3 years)			

* pooled and apheresis platelets, ** estimated 200,000 components transfused annually

**Table 6 pathogens-12-00484-t006:** Summary of HEV seroprevalence (anti-HEV IgG-positive sample rate) and viremia rate in blood donors in Finland compared with data from selected European countries.

Country	Blood Donor Screening	HEV IgG Seroprevalence, %	HEV RNA Positivity Rate (%)	References
Finland	No	7.4	1:5784 (0.017)	This study
Austria	No	13.6	1:8416 (0.012)	[37]
Belgium	No	8.7	1:5448 (0.018)	[36]
Bulgaria	No	25.9	na	[44]
Croatia	No	21.5	na	[45]
Denmark	No	19.8	1:2330 (0.043)	[25,46]
France	Yes	22.4	1:2218 (0.045)	[21,47]
Germany	Yes	29.5	1:1268 (0.079); 1:597 (0.17) *	[20,48]
Ireland	Yes	5.3	1:4997 (0.02)	[19]
Italy	No	8.7	0; 1:157 **	[49,50]
Netherlands	Yes	26.7	1:762 (0.13)	[51,52]
Poland	No	43.5	1:2109 (0.047)	[53]
Spain	Yes ***	20.0	1:4341 (0.023); 1:3333 (0.030)	[24,54]
Sweden	No	17.0	1:7986 (0.01)	[55,56]
Switzerland	Yes	20.4	na	[23]

* Testing in MP96 vs. ID; ** 0/10011 [49] vs. 2/313 [50]; *** Catalonia region.

## Data Availability

The data supporting the reported results can be found at the Finnish Red Cross Blood Service and at the Finnish Institute for Health and Welfare. Sequence data generated in the present study is available in the GenBank with accession numbers OQ596308, OQ596309, and OQ596310.

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
