# Peer review of "Hepatitis E Virus in Finland: Epidemiology and Risk in Blood Donors and in the General Population"

_pathogens, 2023, doi:10.3390/pathogens12030484_

Round 1

Reviewer 1 Report

Review report

This article describes research done on Finnish blood donor population regarding transfusion transmitted HEV infections. This study is interesting and there is a basis of a nice paper here. However, I do suggest certain things, which need attention, improvement and clarification to support and strengthen the overall impact of the article.

Points for attention:

Title: It would be good if the authors would change the title because it is a little bit confusing.  

Abstract:

Authors should include the most relevant conclusions in the Abstract.

Keywords: Good choice of keywords.

Introduction:

It is necessary that the authors explain the purpose, aim and the importance/relevance and novelty of this research.

Lines 38/39: Sentence “Genotypes 1 and 2 infects humans alone, are waterborne and causative for epidemics among humans in developing countries” is too long and unclear.

There are reports available which have shown that in comparison to the general population, a statistically higher seroprevalence is found in pig farmers and veterinarians in comparison to the general population. This suggests that a contact exposure to domestic pigs may also be a risk factor. So, it would be good if the authors would mention that in the introduction, maybe between the lines 43 and 47.

Line 67: The “]” and “.” are missing after the reference.

Line 73: Remove the space before “Data”

Materials and Methods:

Authors should describe sample collection more clearly. Data about samples is listed in lines 89, 136, 140/141, 156 and is difficult to follow.

Line 89: Remove space before “After”.

Line 116: remove space before “Transcription-mediated”

Authors should describe Procleix HEV assay in more details (single-tube, highly sensitive detection of HEV RNA in blood and plasma donations). Also, does this assay implement the positive and negative control for the evaluation of results (describe).

Also, I suggest that the authors describe included positive (is the positive control referent sample or is it isolated in the laboratory, is it sequenced and deposited in GenBank) and negative controls (aliquots of pure water or sample that previously tested negative for USUV).

Results:

Chapter 3.1. describes HEV RNA detection in blood donors. Why are serological results mentioned here?

Authors should indicate from which article they adopted the proposed phylogeny and reference sequences. Authors should also indicate did they include similar sequences obtained using the BLAST algorithm.

Line 239: remove space before “Cases”

Table 5. Please rephrase because the description is not clear enough.

Chapter 3.5. should be removed to the chapter “Conclusions”

Discussion:

The authors should interpret the significance of the results obtained.

Authors should write the significance impact of the study.

Lines 285/286: The sentence “Data on the prevalence in Finnish blood donors has been missing which has hindered assessment of the risk for HEV transmission via blood transfusion” is unclear, please rephrase.

Lines 324, 339, 351, 362, 363, 372, 380 remove spaces.

Line 383: Rephrase “European comparison”

References:

Authors should write references according to the Pathogens style prescribed in the “Instructions for Authors”.

References 45, 47, 48: For thesis: Author, A.B. Title of Thesis, Degree-Graduating University, Location of University, Date of Completion.

Conclusive remarks: This manuscript needs several improvements in the English Style, as well as formatting to improve the readability of the document. Minor revision is needed. Authors should revise carefully chapter “Materials and Methods” because protocols used are not written clearly. Authors should use the same font throughout the manuscript.

Author Response

Response to Reviewer 1 Comments

Comments and Suggestions for Authors

Review report

This article describes research done on Finnish blood donor population regarding transfusion transmitted HEV infections. This study is interesting and there is a basis of a nice paper here. However, I do suggest certain things, which need attention, improvement and clarification to support and strengthen the overall impact of the article.

Points for attention:

Title:

It would be good if the authors would change the title because it is a little bit confusing.  

Response: Title has been revised to improve clarity. Revised title “Hepatitis E Virus in Finland: Epidemiology and Risk in Blood Donors and in the General Population”.

Abstract:

Authors should include the most relevant conclusions in the Abstract.

Response: Conclusion has been added to the abstract (two sentences at the end of the abstract).

Keywords: Good choice of keywords.

Introduction:

It is necessary that the authors explain the purpose, aim and the importance/relevance and novelty of this research.

Response: Last two paragraphs of the introduction has been revised to improve the clarity of the aim and emphasize the relevance and novelty of the study.

Lines 38/39: Sentence “Genotypes 1 and 2 infects humans alone, are waterborne and causative for epidemics among humans in developing countries” is too long and unclear.

Response: Sentence is revised to improve clarity.

There are reports available which have shown that in comparison to the general population, a statistically higher seroprevalence is found in pig farmers and veterinarians in comparison to the general population. This suggests that a contact exposure to domestic pigs may also be a risk factor. So, it would be good if the authors would mention that in the introduction, maybe between the lines 43 and 47.

Response: This information has been added to the introduction as suggested. References Krumbholz et al. and Chaussade et al. have been added accordingly.

Line 67: The “]” and “.” are missing after the reference.

Response: Changed accordingly 

Line 73: Remove the space before “Data”

Response: Changed accordingly

Materials and Methods:

Authors should describe sample collection more clearly. Data about samples is listed in lines 89, 136, 140/141, 156 and is difficult to follow.

Response: Samples collection section has been reorganized and modified to improve clarity (presented in sections 2.1 and 2.2).

Line 89: Remove space before “After”.

Response: Changed accordingly.

Line 116: remove space before “Transcription-mediated”

Response: Changed accordingly

Authors should describe Procleix HEV assay in more details (single-tube, highly sensitive detection of HEV RNA in blood and plasma donations). Also, does this assay implement the positive and negative control for the evaluation of results (describe).

Response: Changed accordingly, more detailed information on the assay has been added to section 2.3.

Also, I suggest that the authors describe included positive (is the positive control referent sample or is it isolated in the laboratory, is it sequenced and deposited in GenBank) and negative controls (aliquots of pure water or sample that previously tested negative for USUV).

Response: Changed accordingly, more detailed information on the assay calibrators/controls has been added to section 2.3.

Results:

Chapter 3.1. describes HEV RNA detection in blood donors. Why are serological results mentioned here?

Response: Revised by transfer of serological results to chapter 3.2.

Authors should indicate from which article they adopted the proposed phylogeny and reference sequences. Authors should also indicate did they include similar sequences obtained using the BLAST algorithm.

Response: Revised by adding description in section 2.4, end of last paragraph and adding reference articles Tamura et al and Saitou et al. 

Line 239: remove space before “Cases”

Response: Changed accordingly

Table 5. Please rephrase because the description is not clear enough.

Response: Table 5 has been revised to improve clarity and readability.

Chapter 3.5. should be removed to the chapter “Conclusions”

Response: A separate chapter 5.“Conclusions” section has been added and content on the chapter 3.5 has been moved to the new section. Table 6 has been removed from this section and transferred before the last pragraph in chapter 4. Discussion.

The authors should interpret the significance of the results obtained.

Authors should write the significance impact of the study.

Response: Significance and impact of the study has been further interpreted by adding a separate conclusions section and including this interpretation in the last paragraph of this new section. Description of the impact of the study is also added to the introduction section.

Lines 285/286: The sentence “Data on the prevalence in Finnish blood donors has been missing which has hindered assessment of the risk for HEV transmission via blood transfusion” is unclear, please rephrase.

Response: Sentence has been revised to improve clarity.

Lines 324, 339, 351, 362, 363, 372, 380 remove spaces.

Response: changed accordingly.

Line 383: Rephrase “European comparison”

Response: Rephrased to “comparison to prevalence in European countries in general”

References:

Authors should write references according to the Pathogens style prescribed in the “Instructions for Authors”.

References 45, 47, 48: For thesis: Author, A.B. Title of Thesis, Degree-Graduating University, Location of University, Date of Completion.

Response: Style of references has been checked and revised to align with the Pathogens style.

Conclusive remarks:

This manuscript needs several improvements in the English Style, as well as formatting to improve the readability of the document. Minor revision is needed. Authors should revise carefully chapter “Materials and Methods” because protocols used are not written clearly. Authors should use the same font throughout the manuscript.

Response: Changes to improve the language and style has been made throughout the manuscript, changes available as mark-up. The font has been checked and aligned.

Reviewer 2 Report

It is an important study that assessed HEV markers in blood donors in Finland. The authors found that 4 samples tested positive for HEV RNA. Genotyping analysis showed that they belonged to HEV gt3 subtype 3c.  Besides the authors performed seroprevalence assay for anti-HEV IgG and anti-HEV IgM.

The study has some advantages: a) large amount of sample assessed  b) using sensitive automated assay for HEV RNA  c) Important data as blood screening in Finland.

However, there several comments on the study in the present form

a) Introduction : the authors should provide some background on HEV in Finland, especially a lot of studies described HEV in Finland either in human or animals.  PMID: 32894411, PMID: 27621202, PMID: 25568927, PMID: 24340718PMID: 23412813.

These studies should be in introduction and discussion section.

b) The authors should take care of the cited references, most of the cited references are taken from reviews, the authors should cite the original article. Such as camel HEV (WOO et al,..), etc.

c) The authors should submit the three sequenced viruses to Genbank and provide the accession number to the manuscript.

d) Did the author test the anti-HEV IgM positive samples with nested PCR that target 493 bp? also what about HEV Ag? The level of HEV Ag is very high in plasma compared to Stool and suggestive of active HEV infection according to EASL.

e) What about the liver function profiles in HEV RNA positive and anti-HEV IgM positive samples.

f) I highly recommend the authors to compare between the isolated HEV strains from humans (this study) and the reported strains in fruit, vegetables, and animals in Finland to track the circulating strains (please see above some references).

Author Response

Response to Reviewer 2 Comments

Comments and Suggestions for Authors

It is an important study that assessed HEV markers in blood donors in Finland. The authors found that 4 samples tested positive for HEV RNA. Genotyping analysis showed that they belonged to HEV gt3 subtype 3c.  Besides the authors performed seroprevalence assay for anti-HEV IgG and anti-HEV IgM.

The study has some advantages: a) large amount of sample assessed  b) using sensitive automated assay for HEV RNA  c) Important data as blood screening in Finland.

However, there several comments on the study in the present form

a) Introduction: the authors should provide some background on HEV in Finland, especially a lot of studies described HEV in Finland either in human or animals.  PMID: 32894411, PMID: 27621202, PMID: 25568927, PMID: 24340718, PMID: 23412813.

These studies should be in introduction and discussion section.

Response: Changed accordingly by adding relevant background information on the studies reported in Finland (references PMID: 32894411, PMID: 27621202, PMID: 25568927, PMID: 24340718 included).

  1. b) The authors should take care of the cited references, most of the cited references are taken from reviews, the authors should cite the original article. Such as camel HEV (WOO et al,..), etc.

Response: Changed by adding references to original articles (e.g. references number 6-9 and 12 in the revised version of the article).

  1. c) The authors should submit the three sequenced viruses to Genbank and provide the accession number to the manuscript.

Response: Sequences have been submitted to GenBank and accession numbers have been added to material & methods in the end of section 2.4.

  1. d) Did the author test the anti-HEV IgM positive samples with nested PCR that target 493 bp? also what about HEV Ag? The level of HEV Ag is very high in plasma compared to Stool and suggestive of active HEV infection according to EASL.

Response: HEV IgM positive samples were not tested with nested PCR (samples were RNA negative) or with HEV Ag test.

  1. e) What about the liver function profiles in HEV RNA positive and anti-HEV IgM positive samples.

Response: Liver function profiles were not included in testing and unfortunately the testing can’t be extended to additional parameters. Also, the ethical approval did not include clinical assessment of HEV RNA positive donors by the Blood Service (the donors were recommended to make appointment with their GP).

  1. f) I highly recommend the authors to compare between the isolated HEV strains from humans (this study) and the reported strains in fruit, vegetables, and animals in Finland to track the circulating strains (please see above some references).

Response: In our study we performed genotyping to evaluate the risk of autochtonous zoonotic hepatitis E. We see that this level of information is sufficient in regard of the aim of our study. Phylogenetic analyses would be interesting and is a topic for future studies.  

Round 2

Reviewer 2 Report

Although the authors did not provide answers to all my questions such as liver parameters and phylogenetic analysis to type the circulating strains in Finland, the manuscript is important and should be published.